# Compressed sensing-based approach identifies modular neural circuitry driving learned pathogen avoidance

Timothy Hallacy[1]*, Abdullah Yonar[2], Niels Ringstad[3], Sharad Ramanathan[2]*

[1]Biophysics Program, Harvard University, Cambridge, United States; [2]Departments of Molecular and Cellular Biology, and of Stem Cell and Regenerative Biology, John A. Paulson School of Engineering and Applied Sciences, Harvard University, Cambridge, United States; [3]Department of Cell Biology, Skirball Institute of Biomolecular Medicine, New York University Grossman School of Medicine, New York, United States

## eLife Assessment

This **important** study describes a neural circuit contributing to two behavioral processes affecting pathogen avoidance in the nematode *C. elegans*. The method used to identify specific contributing neurons is innovative, and the experimental evidence supporting the major claims is **solid**. This study will be of interest to neuroscientists studying behavior, in particular in *C. elegans*.

**\*For correspondence:**
timothyhallacy@gmail.com (TH);
sharad@cgr.harvard.edu (SR)

**Competing interest:** The authors declare that no competing interests exist.

**Abstract** An animal's survival hinges on its ability to integrate past information to modify future behavior. The nematode *Caenorhabditis elegans* adapts its behavior based on prior experiences with pathogen exposure, transitioning from attraction to avoidance of the pathogen. A systematic screen for the neural circuits that integrate the information of previous pathogen exposure to modify behavior has not been feasible because of the lack of tools for neuron type-specific perturbations. We overcame this challenge using methods based on compressed sensing to efficiently determine the roles of individual neuron types in learned avoidance behavior. Our screen revealed that distinct sets of neurons drive exit from lawns of pathogenic bacteria and prevent lawn re-entry. Using calcium imaging of freely behaving animals and optogenetic perturbations, we determined the neural dynamics that regulate one key behavioral transition after infection: stalled re-entry into bacterial lawns. We find that key neuron types govern pathogen lawn-specific stalling but allow the animal to enter nonpathogenic *Escherichia coli* lawns. Our study shows that learned pathogen avoidance requires coordinated transitions in discrete neural circuits and reveals the modular structure of this complex adaptive behavioral response to infection.

## Introduction

Animals use past experiences to modify future behavior, and this behavioral plasticity is essential for an animal's fitness and survival. Previous studies suggest that sparse sets of neurons, such as those in the mushroom bodies of flies (*Zhao et al., 2021*, *de Belle and Heisenberg, 1994*, *Pascual and Préat, 2001*, *Heisenberg, 2003*) or in the human hippocampus (*Vargha-Khadem et al., 1997*, *Bird and Burgess, 2008*, *Kovács, 2020*, *Duff et al., 2019*, *FeldmanHall et al., 2021*), can play pivotal roles in integrating information from prior experiences and using it to influence subsequent behavior. These critical neurons at bottlenecks of neural networks make up a small fraction of the total size of the nervous system. Identification of these key neurons represents a crucial step towards understanding

both the neural and molecular mechanisms responsible for encoding experiences and modulating behavior.

Despite its small nervous system, the nematode *Caenorhabditis elegans* displays a robust capacity to modify its behavior based on experience (*Gourgou et al., 2021*, *Amano and Maruyama, 2011*, *Eliezer et al., 2019*, *Zhang and Zhang, 2012*, *Ardiel and Rankin, 2010*). One example of this is learned pathogen avoidance. Worms are initially attracted to pathogenic bacteria such as *Pseudomonas aeruginosa* (strain PA14). However, after several hours of exposure, animals associate infection with PA14 specific cues and change their behavior to avoid these bacteria (*Meisel and Kim, 2014*, *Zhang et al., 2005*, *Chen et al., 2013*, *Shivers et al., 2009*, *Singh and Aballay, 2019*, *Kaletsky et al., 2020*). This behavioral transition reduces the chance of infection and thus increases the worm's odds of survival (*Shivers et al., 2009*, *Kim et al., 2002*, *Tan et al., 1999*, *Reddy et al., 2009*). How does the worm's nervous system encode this pathogen experience and use this information to change behavior? Previous studies uncovered a range of sensory cues that govern the worm's behavioral transition, ranging from chemosensory cues (*Meisel et al., 2014*, *Hao et al., 2018*, *Ha et al., 2010*, *Harris et al., 2014*) and the worm's innate immune response (*Zhang et al., 2005*, *Lee and Mylonakis, 2017*) to mechanosensory inputs arising from the bacteria's biofilm (*Reddy et al., 2011*, *Chang et al., 2011*, *Bai et al., 2021*). Olfactory stimuli have also been implicated in learned pathogen avoidance through serotonin modulation (*Shivers et al., 2009*, *Zhang et al., 2005*). The underlying neural circuits that encode information about past experiences and their dynamics remain unknown despite this. One strategy to find such neurons would be to carry out a comprehensive screen of the nervous system. However, such a screen is technically challenging due to the lack of neuron subtype-specific promoters and the necessity of systematic timed perturbations of neural activity during pathogen exposure to disrupt the animal's ability to learn to avoid pathogens in the future.

To discover neurons whose activity patterns governed experience-dependent pathogen avoidance, we performed a comprehensive screen of the nervous system using a compressed sensing-based approach (*Lee et al., 2019*) that overcomes these technical challenges. Through this screen, we discovered that the behavioral transition from attraction to avoidance of PA14 involves transitions in two subbehaviors - exit from the bacterial lawn and lawn re-entry. We found that distinct sets of neurons regulate each subbehavior. In particular, the aversion to re-entry is controlled by two neuronal types, AIY and SIA, which encode pathogen exposure through sustained downregulation of neural activity. Our data further indicated that AIY regulates PA14-specific aversion and drives stalling at the edge of the pathogenic lawn. Finally, we used the identification of neural substrates of learned pathogen avoidance to explore how long-term changes in neural dynamics might be mediated by neuropeptide signaling between critical neurons that govern transitions in this behavior.

## Results

### Prior pathogen exposure alters two behavioral modules to generate learned pathogen avoidance

Infection of the nematode *C. elegans* by pathogenic bacteria elicits learned pathogen avoidance behavior. Naïve animals dwell and feed on lawns of pathogenic *P. aeruginosa* (PA14), but after infection, animals avoid *P. aeruginosa* lawns (*Meisel and Kim, 2014*, *Reddy et al., 2011*). To obtain high-resolution measurements of learned pathogen avoidance behavior, we monitored animals that had been placed on a lawn of pathogenic PA14 for 18 hours. We counted the fraction of animals that remained on the lawn over time (*Figure 1A*) at 30-minute intervals. Within 8 hours, half of the animals changed their foraging behavior and left the lawn of pathogenic bacteria. By contrast, nearly all the animals placed on a control lawn of non-pathogenic *Escherichia coli* (OP50) bacteria remained on the lawn during 18 hours of monitoring (*Figure 1A*).

The fractional occupancy of worms on a bacterial lawn is affected in principle by two processes: lawn exit and lawn re-entry. We determined how PA14 exposure affected each of these processes. After four hours in the presence of pathogen, the rate constant for the lawn exit (exits per animal per hour) increased dramatically from zero and reached a plateau of roughly one exit event per worm-hour after 10 hours (*Figure 1B*). We observed that worms that left lawns of pathogen after 10–16 hours of exposure repeatedly attempted to return to the lawn but stalled for long periods upon contact with the lawn edge (*Figure 1C*, *Video 1*). We quantified this re-entry defect by measuring the latency to re-entry, that

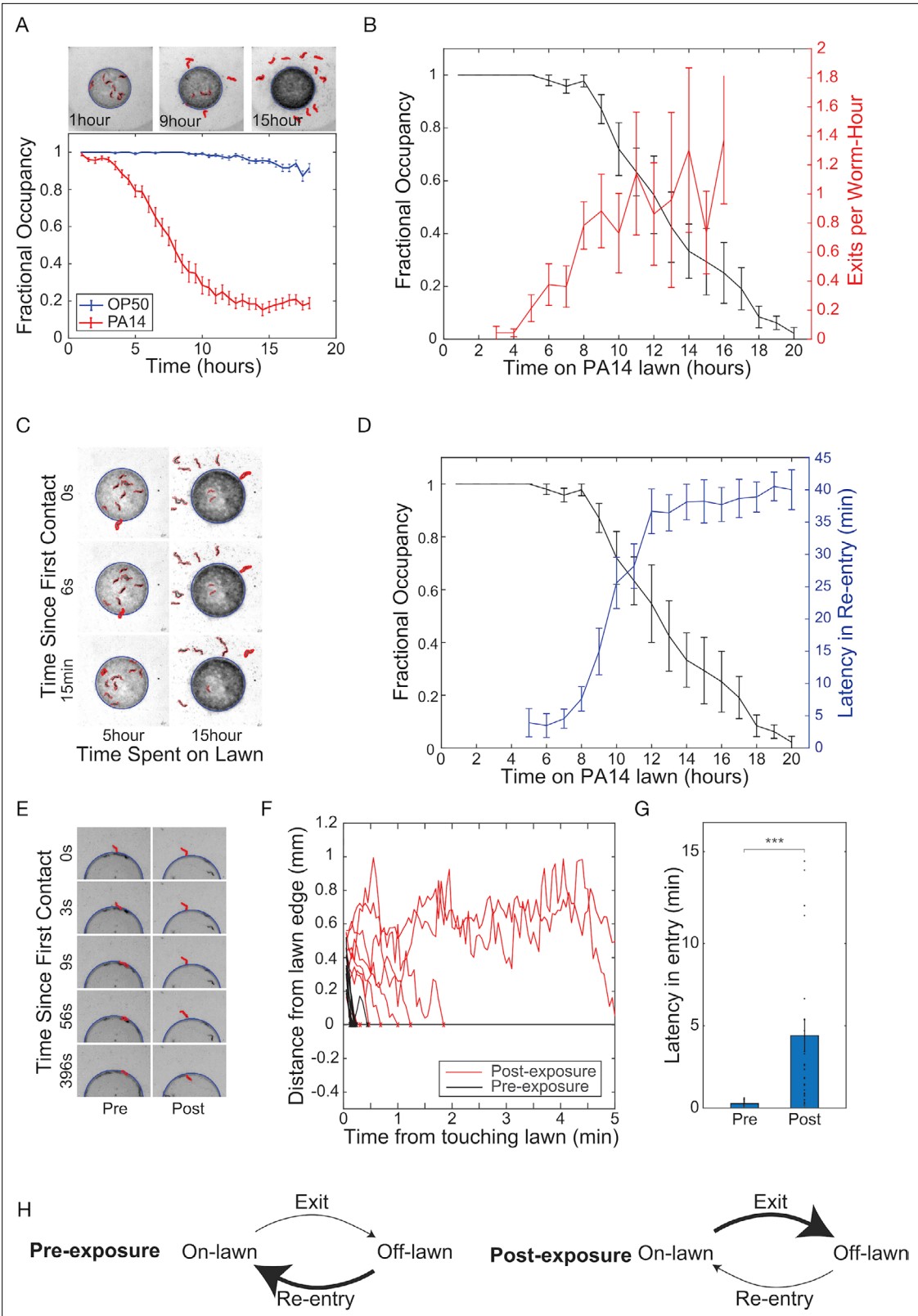

**Figure 1.** Exposure to pathogenic bacteria inhibits the ability of worms to re-enter the bacteria lawn. (**A**) *Above*: representative sequence of phase contrast images of *P. aeruginosa* PA14 lawn evacuation at t=1, 9, and 15 hours after deposition of *C. elegans* onto lawn. Evacuated worms are highlighted in red using imaging processing. *Below*: fractional occupancy of *C. elegans* on bacteria lawns plotted against time spent on pathogenic PA14 lawns (red) or non-pathogenic *E. coli* OP50 lawns (blue). Mean occupancy and SEM over n=25 independent experiments with 10 worms each. (**B**)

*Figure 1 continued on next page*

*Figure 1 continued*

*C. elegans* fractional occupancy (black) and the rate constant of *C. elegans* exit (exits events/hour/animal, red) plotted against time spent on pathogenic PA14 lawns. The exit rate constant increased with time spent on the lawn. Mean and SEM over n=5 samples of 10 worms each. (**C**) Representative phase contrast images of a worm attempting to re-enter PA14 lawn after 5 hours (left column) and 15 hours (right column) of PA14 exposure. Time labels denote time after first contact of worm with bacteria lawn. (**D**) Latency in re-entry (delay in the re-entry of the animal onto the bacteria lawn upon first contact with the lawn, red) and fractional occupancy (black) plotted against duration of exposure for worms on PA14 lawns. Mean and SEM over n=5 samples of 10 worms each. (**E**) Representative sequence of phase contrast images taken from a fresh lawn assay experiment to investigate worm entry dynamics. Worms (highlighted in red) are transferred to a fresh lawn of PA14 and their entry dynamics are measured pre-PA14 exposure (left) and post-15 hours of PA14 exposure (right). Time labels denote time after first contact of worm with bacteria lawn. (**F**) Representative plots of distance of center of mass of each animal from the edge of the PA14 lawn versus time after first contact with lawn, pre (black, n=11) and post 15 hours of PA14 exposure (red, n=9). (**G**) Latency of entry of worms pre (n=35) and 15 hours post (n=29) PA14 exposure. Post-exposed worms increased latency of entry tenfold compared to pre-exposed control. (**H**) Model for lawn evacuation. Pre-exposed worms (left) have high re-entry rates and low exit rates. Exposure to pathogenic bacteria causes a transition in behavior (right), where worms increase exit rate and decrease re-entry rate, leading to net lawn evacuation.

The online version of this article includes the following figure supplement(s) for figure 1:

**Figure supplement 1.** Lawn evacuation following neural inhibition.

is, the delay between first contact with the lawn and re-entry of the animal into the lawn. Animals that had been exposed to a pathogen for short periods rapidly re-entered the lawn after an exit with latencies of only a few seconds (*Figure 1D*). After 5 hours of exposure to pathogen, the latency of re-entry began to increase, and after 12 hours of exposure, the mean latency to re-entry was 36.7±3.5 minutes (*Figure 1D*, *Videos 2 and 3*). Increases in lawn-leaving rates and latency to re-entry were correlated and coincided with the observed evacuation of the lawn of pathogenic bacteria (*Figure 1D*).

To demonstrate that the experience of pathogen exposure was driving the observed changes in behavior, as opposed to some change in the bacterial lawn caused by foraging animals, we performed an assay to test worm entry dynamics on a fresh lawn of PA14. We exposed animals to pathogenic PA14 for 15 hours, collected those that had left the lawn, and then compared the latency of lawn entry of this cohort to the latencies of a cohort of naive animals that had not experienced pathogen. Naive worms rapidly entered a fresh lawn of pathogen, whereas worms that had experienced 15 hours of pathogen exposure displayed increased latency of entry into such fresh lawns (*Figure 1E–G*). The results of this fresh lawn assay indicated that pathogen exposure changed the internal state of the worm to drive the observed behavioral transition from re-entering the lawn rapidly to stalled re-entry (*Figure 1H*).

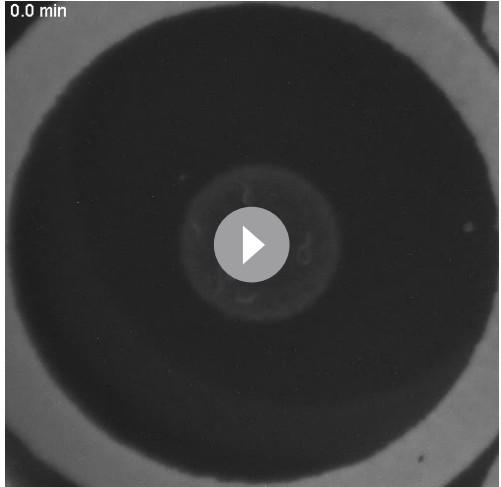

**Video 1.** Time-series video of a lawn evacuation assay taken 2 hours into the assay.

https://elifesciences.org/articles/97340/figures#video1

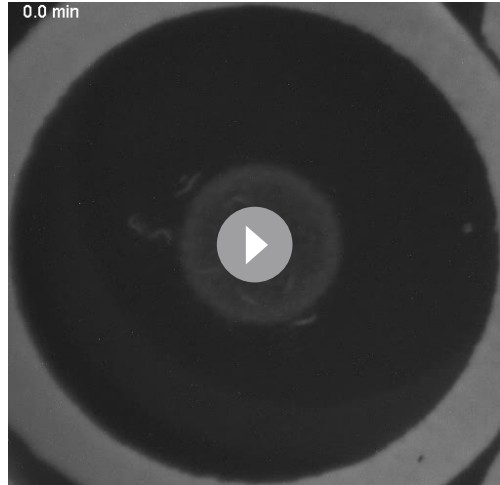

**Video 2.** Time-series video of a lawn evacuation assay taken 8 hours into the assay.

https://elifesciences.org/articles/97340/figures#video2

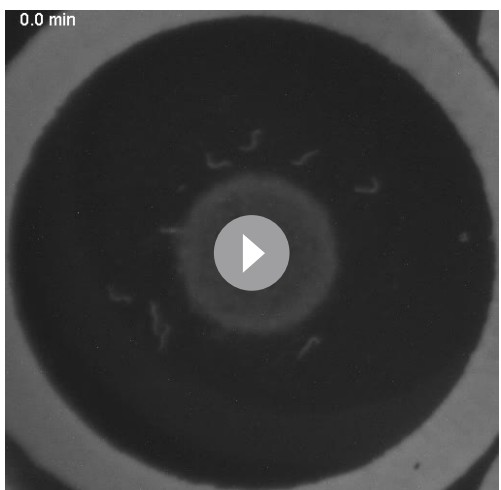

0.0 min

**Video 3.** Time-series video of a lawn evacuation assay taken 14 hours into the assay.

https://elifesciences.org/articles/97340/figures#video3

## A compressed sensing-based optogenetic screen to identify neurons that function in learned pathogen avoidance

We next sought to identify neurons that regulate interactions of *C. elegans* with lawns of pathogenic *P. aeruginosa*. Conventional approaches to identifying neurons required for a *C. elegans* behavior involve targeting individual neuron types by microablation (*Avery and Horvitz, 1989*, *Gray et al., 2005*, *Kobayashi et al., 2013*) or optogenetics (*Liu et al., 2022a*, *López-Cruz et al., 2019*, *Sordillo and Bargmann, 2021*). These approaches can be time-consuming, and applying them to perform unbiased screens of the entire nervous system can require the generation of large numbers of transgenic lines.

We previously showed that an optogenetic screen for neurons that drive a behavior can be performed more efficiently using multiplexed optogenetic manipulations of neurons followed by a compressed sensing analysis to infer individual key neuron types (*Lee et al., 2019*, *Candes and Wakin, 2008*, *Donoho, 2006*). We performed a compressed sensing-based screen using a panel of 29 transgenic *C. elegans* lines, each expressing the light-gated ion channel Archaerhodopsin-3 (Arch3) under a different promoter (*Supplementary file 1*). The panel used for this study focused on interneurons, and covered 54 classes of interneuron, 25 classes of sensory neuron, and 8 classes of motor neuron (*Hobert et al., 2016*; *White et al., 1986*; *Xu et al., 2013*). We optically inhibited neurons during the early stages of pathogen exposure when animals presumably associate pathogen-specific cues with sickness. We then monitored the subsequent dynamics of lawn leaving and re-entry. Animals were transferred to pathogenic lawns, given 1 hour to settle, and then illuminated for 2 hours with pulses of 525 nm green light (1 second on/off, 5 mW/mm$^2$) to inhibit neurons expressing Arch3 in that line (*Figure 2A*). The behavior of these animals was then recorded for a total of 18 hours at 3-minute intervals (*Figure 2A*). We performed the same measurements of matched controls that had not been fed the opsin cofactor all-trans retinal and were thus insensitive to photoinhibition (*Figure 2A*). To quantify the effect of optogenetic inhibition on behavior, we calculated for each strain a differential retention index—the difference in the area between the temporal lawn occupancy curves of transgenic animals exposed to inhibitory light and a paired no-ATR control (*Figure 2B*). Six strains showed significant changes in differential retention index after neural silencing compared to controls (*Figure 2B*). Silencing neurons in these strains during the first 2 hours of pathogen exposure augmented lawn-leaving over the next 18 hours (*Figure 1—figure supplement 1A*).

We next asked whether the accelerated pathogen avoidance observed upon neural silencing resulted from increased lawn exits, increased latency to re-entry, or both. Two Archaerhodopsin lines (P*dop-2::Arch3* and P*mpz-1::Arch3*) showed increased exit rates, while four lines (P*flp-4::Arch3*, P*sams-5::Arch3*, P*ttx-3::Arch3*, and P*npr-4::Arch3*) did not show changes in exit rates upon inhibition (*Figure 2C and D*). Neural inhibition that increased lawn exit rates dramatically increased the number of tracks outside of the lawn (*Figure 2E*). We next measured the effects of the early neural silencing on lawn re-entry. We found that four lines (P*dop-2::Arch3*, P*npr-4::Arch3*, P*sams-5::Arch3*, and P*ttx-3::Arch3*) showed significant increases in latency to re-entry in response to activation of Archaerhodopsin (*Figure 2F and G*). For example, the inhibition of the P*npr-4::Arch3* line over the first 2 hours of the experiment dramatically decreased lawn re-entry over the entire time course, resulting in worm trajectories stalling at the lawn edge (*Figure 2H*). Inhibition of neurons on non-pathogenic OP50 produced dramatically different effects on both exit and entry compared to PA14 (*Figure 1B and C*). In particular, none of the optogenetic lines showed changes in latency to re-entry on OP50 (*Figure 1C*), suggesting that this effect is specific to PA14 exposure. These results indicated that

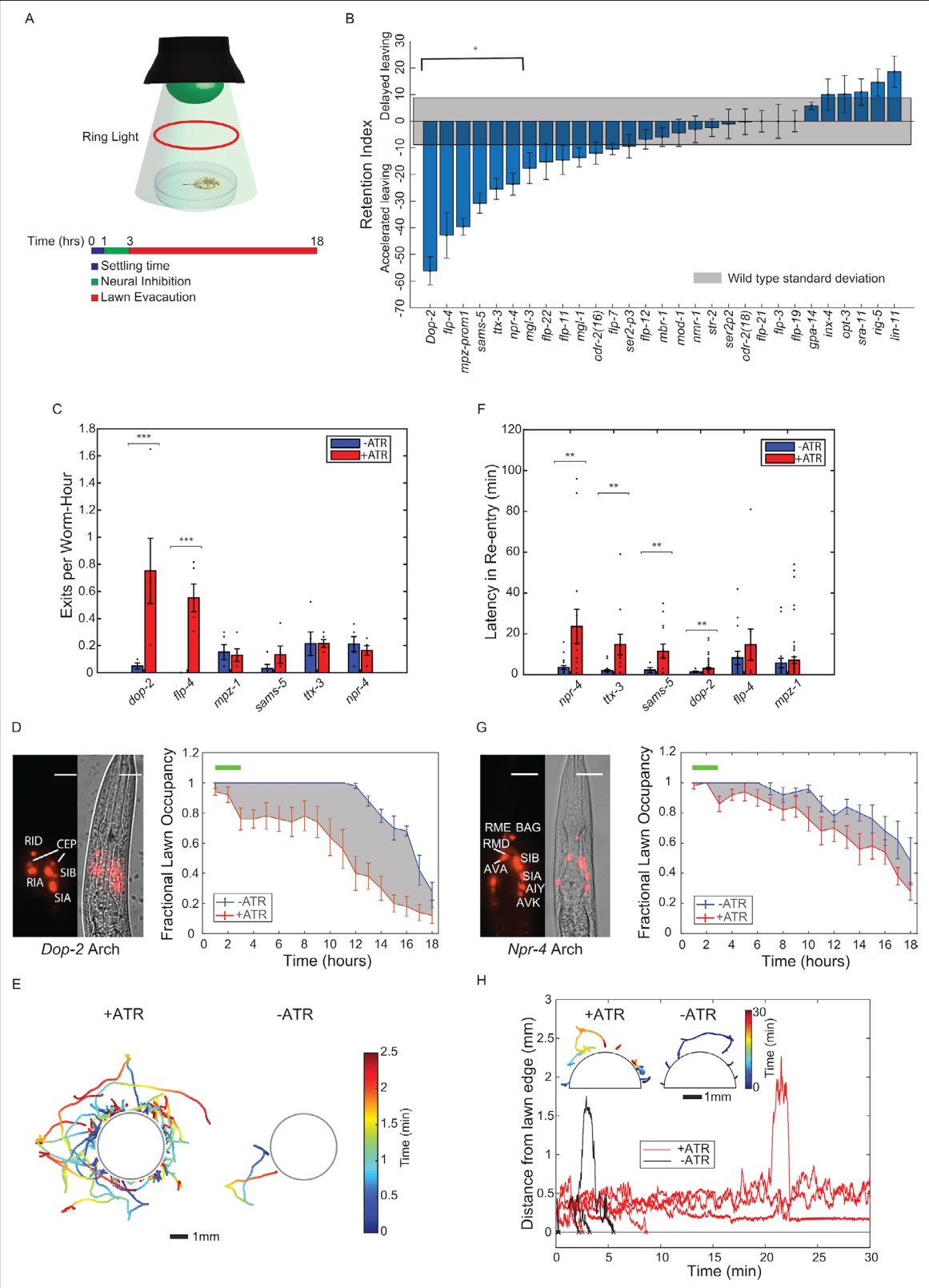

**Figure 2.** An optogenetic screen of the *C. elegans* nervous system to identify neurons controlling exit and re-entry into pathogenic bacteria lawns. (**A**) Experimental protocol used to identify neurons that modulate long-term changes in PA14 lawn evacuation dynamics. In each experiment, animals of one transgenic line expressing archaerhodopsin were placed onto a lawn of PA14 and allowed to settle for 1 hour (blue segment, in schematic). Worms were illuminated with green light (525 nm) to inhibit activity for 2 hours in neurons expressing archaerhodopsin (green segment, in schematic). Worm

*Figure 2 continued on next page*

*Figure 2 continued*

evacuation was monitored over an additional 15 hours (red time segment, schematic). Worms are illuminated for imaging with a red ring light. (**B**) Differential retention metric for 29 transgenic lines. The differential retention metric for each transgenic line was determined by measuring the difference in the area under lawn evacuation curves with and without neural inhibition (see 'Materials and methods'). Differential retention metrics were compared against the standard deviation in retention metrics from uninhibited control worms (gray) to determine statistical significance (see methods). 6 of the 29 lines (*dop-2, flp-4, mpz-prom1, sams-5, ttx-3,* and *npr-4*) showed statistically significant differential retention metrics. Mean and SEM over n=5 samples of 10 worms each. (**C**) Rate constant of exit (exit events/hour/animal) for statistically significant transgenic lines (**B**). Two lines showed statistically significant changes in rate constant (*dop-2* and *flp-4*) between inhibition (red,+ATR) and no inhibition (blue, -ATR) control. Mean and SEM over n=5 samples of 10 worms each. (**D**) Left: fluorescence and phase image of P*dop-2*::ARCH-mCherry. Scale bar 15 μm. Right: fractional lawn occupancy of P*dop-2*::ARCH-mCherry as a function of time (right) with (red,+ATR) and without (blue, -ATR) neural inhibition. (**E**) Trajectories of worms exiting the lawn either with (left) or without (right) inhibition of neurons expressing dop-2. Trajectories are taken during the 2-hour timescale of inhibition, and times are taken from the time of lawn exit. Trajectories are color-coded by time and are over n=2 samples of 10 worms each. (**F**) Latency in re-entry (time of re-entry of the animal onto the bacteria lawn after first contact) of statistically significant transgenic lines (**B**). Four lines show statistically significant differences in latency in re-entry (*dop-2, sams-5, ttx-3,* and *npr-4*) between inhibition (red, +ATR) and no inhibition (blue, -ATR) control. (**G**) Representative fluorescence and phase image (left) of P*npr-4*::ARCH-mCherry. Scale bar 15 μm. Fractional lawn occupancy of this line as a function of time (right) with (red,+ATR) and without (blue, -ATR) neural inhibition. (**H**) Representative plots of the distance of individual P*npr-4*::ARCH-mCherry worms from the edge of the lawn with (red, n=5) and without (black, n=5) neural inhibition. Inset: representative trajectories (n=5) color coded by time of P*npr-4*::ARCH-mCherry worms attempting to enter bacteria lawn with (left) and without (right) neural inhibition. All trajectories are taken from lawn exit to re-entry.

The online version of this article includes the following figure supplement(s) for figure 2:

**Figure supplement 1.** Compressed sensing solutions with sparsity parameters for lawn exit and entry.

**Figure supplement 2.** Validation of compressed sensing solutions via Arch expression efficiency through measurement matrix corruption.

**Figure supplement 3.** Validation of robustness of compressed sensing solutions to choice of measurements through promoter removal.

**Figure supplement 4.** Recovery and false positive rate for measurement matrix of choice.

inhibition of neurons during an early phase of learned pathogen avoidance could cause long-term changes in aversion to pathogenic bacteria through modulating distinct behaviors.

## A sparse set of neurons influences the encoding of the memory of pathogen exposure

We next sought to identify specific neurons that influence learned pathogen avoidance. To identify neurons that control specific behaviors (*Figure 3A*), one usually thinks of perturbing $N$ neurons of the nervous system one at a time. These measurements can be visualized as an $N \times N$ set of equations, which can be easily solved to give the relative contribution of each neuron to the phenotype (*Figure 3B*). This approach requires as many measurements as the number of neurons being characterized. Utilizing a compressed sensing-based approach, we can instead formulate our optogenetic screen results as an underdetermined set of equations $\mathbf{M}\vec{w} = \vec{P}$. The matrix, $\mathbf{M}$, is an incoherent measurement matrix of size 29 by 87 measurement matrix (*Figure 3C*), with each row corresponding to each Archaerhodopsin line (experiments were performed on 29 lines) and each column corresponding to neural identity. The matrix element, $\mathbf{M}_{ij}$ is equal to 1 if the $i$th line drives expression in neuron $j$ and is 0 otherwise. $\vec{w}$ corresponds to neural weights, that is, how much each neuron contributes to the phenotype, and $\vec{P}$ is the phenotype vector. The relative contributions (weights) of 87 neuron types to a phenotype can be determined using an L1 norm from these 29 measurements.

We evaluated the phenotype vector $\vec{P}$ for the two behaviors of interest by quantifying the changes to the dynamics of re-entry and exit from the lawns of pathogen. For both exit and re-entry, we calculated differences between optogenetically inhibited and non-inhibited worms for each transgenic line that showed statistically significant effects of inhibition. For lawn exit, the phenotype was the increase in the rate constant of exit due to neural inhibition. For re-entry, the phenotype was the difference between the latency in re-entry caused by neural inhibition. Lines that did not show significant behavioral changes in response to inhibition were assigned a phenotype value of zero.

We inferred neural weights $\vec{w}$ from our matrix equation using Lasso regression (*Tibshirani, 2011*, *Tibshirani, 1996*), which allowed us to solve the underdetermined set of linear equations $\mathbf{M}\vec{w} = \vec{P}$ while simultaneously imposing sparsity constraints on the weight vector by minimizing the sum of the mean squared error $\chi^2 = \left(\mathbf{M}\vec{w} - \vec{P}\right)^2$ and the L1 norm of the solutions, thus minimizing $\left(\mathbf{M}\vec{w} - \vec{P}\right)^2 + \lambda \left\|\vec{w}\right\|_1$ where $\lambda$ is the sparsity parameter. Using this approach, we inferred a set of candidate neurons for

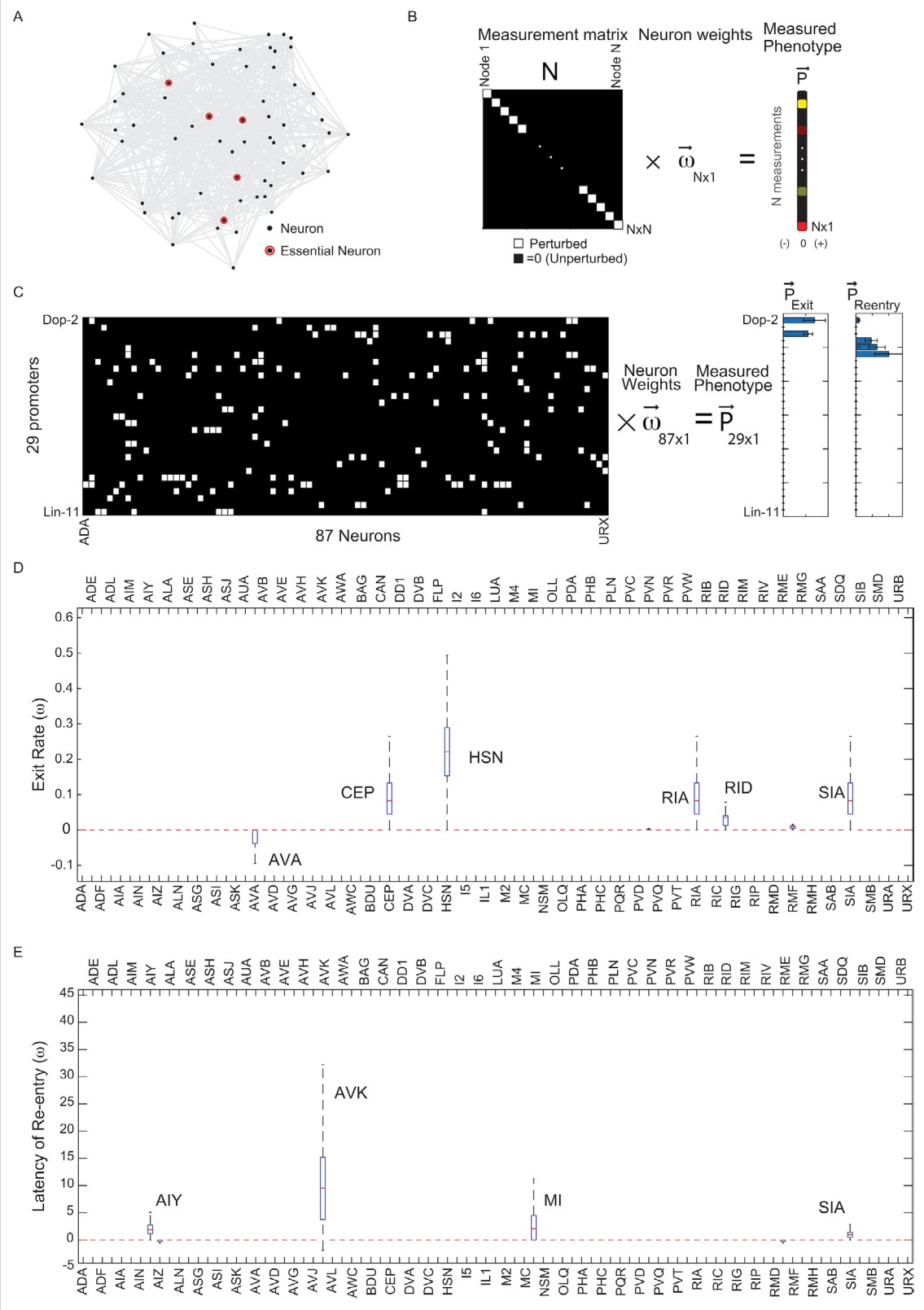

**Figure 3.** Compressed sensing analysis identified a set of neurons controlling worm re-entry and exit. (**A**) Example of a neural network where behavior is controlled by a small number of key nodes (highlighted in red) that make up a small fraction of the total number of nodes (gray). (**B**) Single neuron perturbations can be framed as solving a N by N matrix equation to find the neural contributions to phenotypes. Each diagonal entry (white) corresponds to a single perturbation. Interrogating the entire nervous system to determine the contribution (weight, w) of each neuron to the behavior

*Figure 3 continued on next page*

*Figure 3 continued*

requires as many measurements as the number of neurons in the nervous system (N). (**C**) Using compressed sensing to determine neurons controlling entry and exit from the optogenetic screen by determining their weights (w) from an underdetermined set of equations. These equations can be represented as a 29 × 87 measurement matrix M (left). Rows are promoters and columns are neuron types. Matrix has an entry 1 (white) if the promoter drives expression in that neuron type, else 0 (black). Phenotype vectors $P_{exit}$ and $P_{entry}$ were obtained by taking the difference between the re-entry timescale or rate constant of exit with and without neural inhibition (see 'Materials and methods'). From measurements of the phenotype of 29 lines ($P_{exit}$ and $P_{entry}$), the weights of each of the 87 neurons (w) can be determined using compressed sensing (see 'Materials and methods') (**D**) Median neuron weight contributions to the exit phenotype from 10,000 lasso regression solutions from bootstrapping (see 'Materials and methods'). Neurons with significant weights contributing to the exit phenotype (AVA, CEP, HSN, RIA, RID, SIA) are highlighted. (**E**) Median neuron weight contributions to the re-entry phenotype from 10,000 lasso regression solutions from bootstrapping (see 'Materials and methods'). Neurons with significant weights contributing to the re-entry phenotype (AIY, SIA, AVK, MI) are highlighted.

lawn exit and lawn re-entry (*Figure 3D and E*). We focused primarily on neurons governing lawn re-entry (*Figure 1F*). For lawn re-entry, four key neurons were inferred over a wide range of sparsity parameters: AVK, SIA, AIY, and MI (*Figure 2—figure supplement 1*), with one example solution taken at a point where chi-squared error began to increase (*Figure 3D*). Some neurons were assigned negative weights by this analysis (suggesting that their inhibition promotes lawn re-entry). However, the contributions of these neurons decreased as the sparsity parameter increased, suggesting that these neurons were less important (*Figure 2—figure supplement 1*).

Compressed-sensing analysis of lawn-exit behavior identified contributions from six neuron classes to this behavior: CEPs, HSNs, RIAs, RIDs, and SIAs. Some of these neurons have previously been implicated in lawn retention or dwelling. RIAs are required for learned avoidance of *Pseudomonas* lawns (*Liu et al., 2022b*, *Ha et al., 2010*). CEPs and HSNs are also known to promote dwelling on bacterial lawns through release of dopamine and serotonin, respectively (*Sawin et al., 2000*). Notably, this analysis suggested that the neural circuit governing lawn exits is distinct from the neural circuit governing re-entry.

We next performed several tests to determine the robustness of our solutions, focusing on lawn re-entry behavior. To determine whether variation in archaerhodopsin expression might affect identification of neurons that govern this behavior, we tested the solutions to corrupted versions of the measurement matrix. The four neurons AVK, SIA, AIY, and MI were robustly identified regardless of matrix corruption (*Figure 2—figure supplement 2*). We next tested whether random removal of promoters would alter our solutions, *i.e.*, whether a small subset of strains was driving the identification of neurons. Neuron identification was robust to removal of up to 5 of the 29 promoters (*Figure 2—figure supplement 3*). Finally, we determine false-positive and false-negative rates for compressed sensing-based inference (*Figure 2—figure supplement 4A*), as well as the recovery rate and true positive rate for each of the four neurons (AVK, SIA, AIY, and MI) (*Figure 2—figure supplement 4B–E*). While all four neurons identified have high recovery rates, MI, SIA, and AVK can have true recovery rates below 50%. Thus, we would expect at least one of these neurons to be a false positive. To directly test and determine the roles of the neurons implicated by compressed sensing, we next focused on measuring the activities from these neuron types in freely moving animals.

## Neurons identified by compressed sensing encode the experience of pathogen exposure as a reduction in neural activity

We measured calcium signals in three neuron subtypes (AVK, SIA, and AIY) in unrestrained worms before and after exposure to pathogenic *Pseudomonas*. We excluded MI from this analysis because of its role as a pharyngeal motor neuron; perturbation of MI might have affected lawn re-entry by modulating bacteria ingestion (*Vidal et al., 2022*).

To accurately measure calcium dynamics in AVK, SIA, and AIY, we used a custom real-time image-stabilization microscope capable of tracking and measuring neural dynamics in freely moving worms (*Figure 4A*). The microscope can track worms with 1 µm precision in all three dimensions while performing rotational stabilization by tracking a marker neuron (AWC$^{on}$, P*str-2::mKO*). We demonstrated the capability of this system by imaging neural activity in freely moving *C. elegans* at high magnification for up to an hour without affecting animal behavior (*Lee et al., 2019*).

We tracked neurons in transgenic animals expressing the calcium sensor GCaMP6s in AVK, SIA, or AIY interneurons. Naive animals (pre-pathogen exposure) were imaged on an empty agar plate for

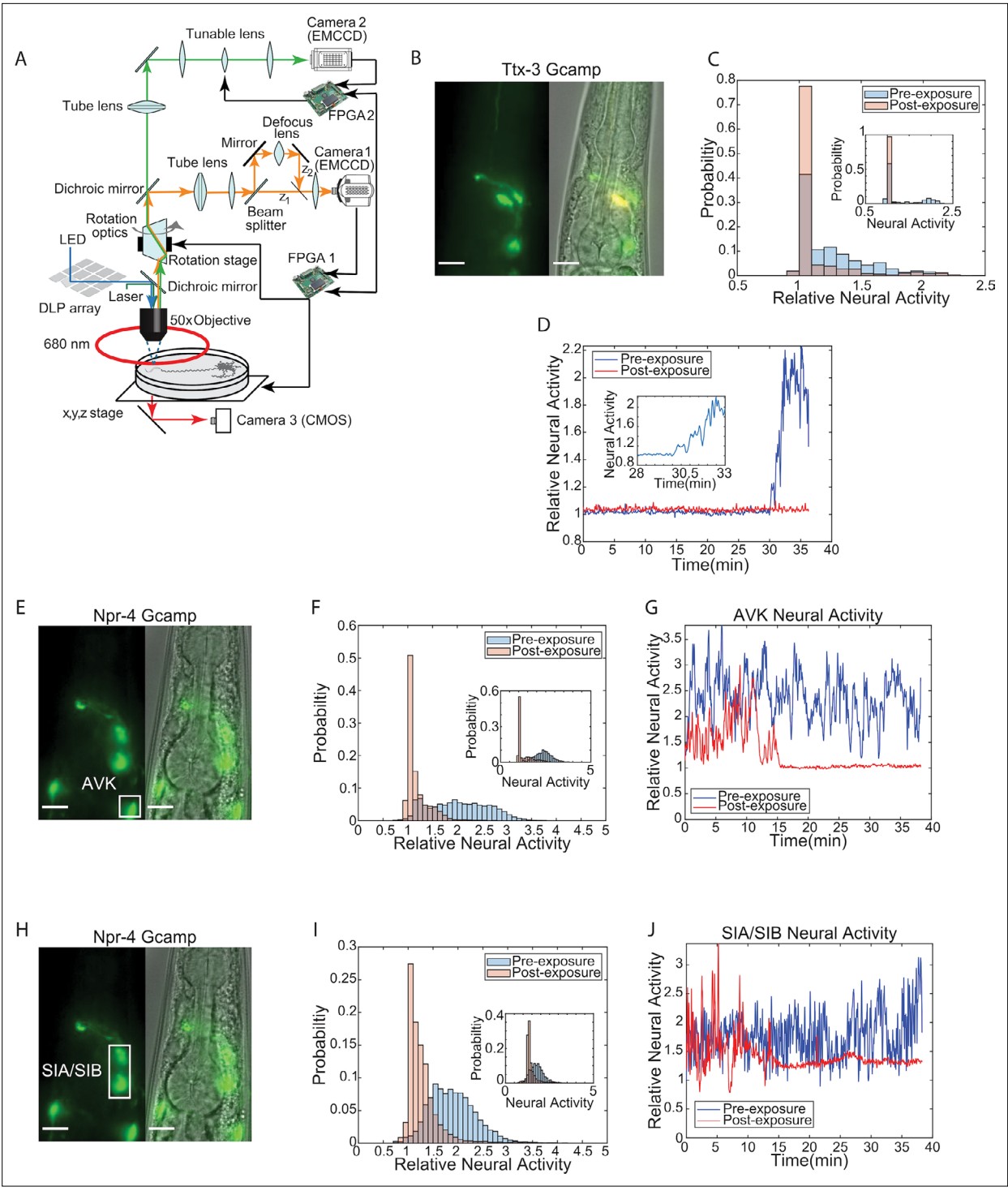

**Figure 4.** Identified neurons responsible for re-entry show reduction in calcium activity following PA14 exposure. (**A**) Schematic of tracking and image stabilization microscope allowing for simultaneous measurement of GCaMP activity and worm position with 1 μm precision (orange: mKO emission, light green: GCaMP emission, blue: GCaMP excitation, dark green: mKO excitation). An mkOrange labeled marker neuron is imaged through Camera 1. This imaging data is transmitted to FPGA 1 for processing. Processed position information is used to control the x, y, z (stage) and rotational optics to stabilize the worm within the field of view. GCaMP imaging data is acquired via Camera 2. A tunable lens being controlled by FPGA 2 scans through the worm to allow for acquisition of GCaMP images at multiple focal planes. A DLP mirror array, controlled by a PC, is used to target light on specific neurons through structured illumination. (**B**) Fluorescent and phase image of P*ttx-3*::GCaMP line used to image AIY neural activity. Scale bar 10 μm. (**C**) Histogram of AIY neural activity pre (blue) and post (red) 24 hours of PA14 exposure normalized to pre-exposure baseline (see 'Materials and methods'). Data taken from n=4 worms over 36 minutes. Inset: histogram of neural activity from a single worm pre (blue) and post (red) 24-hour PA14 exposure.

*Figure 4 continued on next page*

*Figure 4 continued*

(**D**) AIY neural activity over 36 minutes for a single worm pre (blue) and post (red) 24 hours of PA14 exposure normalized to pre-exposure baseline (see 'Materials and methods'). Inset: zoom in of AIY neural activity between 28–33.5 minutes to illustrate neural activity transition in naïve (pre-exposed) worms. (**E**) Fluorescent and phase image of P*npr-4*::GCaMP line used for AVK imaging. Scale bar 10 μm. (**F**) Histogram of AVK neural activity pre (blue) and post (red) 24 hours of PA14 exposure normalized to pre-exposure baseline (see 'Materials and methods'). Data taken from n=7 worms over 36 min. Inset: histogram of neural activity from a single worm pre (blue) and post (red) 24-hour PA14 exposure. (**G**) AVK neural activity for a single worm pre (blue) and post (red) 24 hours of PA14 exposure as a function of time normalized to pre-exposure baseline (see 'Materials and methods'). (**H**) Fluorescent and phase image of P*npr-4*::GCaMP used to image SIA neural activity. Scale bar 10 μm. (**I**) Histogram of AVK neural activity pre (blue) and post (red) 24 hours of PA14 exposure normalized to pre-exposure baseline (see 'Materials and methods'). Data was taken from n=8 worms over 36 min. Inset: histogram of neural activity from a single worm (blue) and post (red) 24-hour PA14 exposure. (**J**) SIA neural activity for a single worm pre (blue) and post (red) 24 hours of PA14 exposure as a function of time normalized to pre-exposure baseline (see 'Materials and methods').

approximately 40 minutes. These worms were then placed onto a PA14 lawn for 24 hours, recovered to an empty assay plate, and imaged for approximately 40 minutes to determine how pathogen exposure affected neural activity. P*npr-4*::GCaMP6s was utilized to measure the activity of AVKs and SIAs, and P*ttx-3*::GCaMP6s was used to measure the activity of AIYs. We found that AIY (***Figure 4C and D***), AVK (***Figure 4F and G***), and SIA (***Figure 4I and J***) all showed reduced neural activity after pathogen exposure. These results were evident both in analyses of populations of animals (***Figure 4C, F and I***), but were also clearly observed in individual worms (***Figure 4C, F and I*** inset and ***Figure 4D, G and J***). Pathogen exposure had different effects on different interneurons. AIY neurons of naive animals showed a significant increase in calcium approximately 30 minutes after transfer to assay plates (***Figure 4D***). By contrast, AIY neurons of animals that had been exposed to pathogens remained quiescent. AVK and SIA neurons of naive animals displayed continuous high-frequency calcium signals. Post-exposure, AVKs and SIAs displayed long periods of quiescence. This data indicated that the activity of neurons that regulate a behavior critical for learned pathogen avoidance is strongly affected by exposure to pathogen. These neurons thus appear to encode the history of exposure to pathogenic bacteria in their neural activity state.

## Modulation of candidate neurons validates their role in inhibition of pathogen lawn re-entry

We next tested how manipulating the activity of neurons identified by compressed sensing as regulators of lawn re-entry affects how naïve animals interact with bacterial lawns. We inhibited each of the three candidate neuron types using Archaerhodopsin and measured how acute neuronal inhibitions affected re-entry into a lawn of pathogen. We performed parallel measurements of a control set of worms that had not been treated with the opsin cofactor ATR. To target specific neurons, we used a DLP mirror array to restrict illumination to cells of interest as previously described (***Lee et al., 2019***).

We found that acute inhibition of AIY in naive animals (no prior PA14 exposure) increased the latency of re-entry onto a fresh lawn of pathogenic *Pseudomonas*, mimicking the effect of prior pathogen exposure (***Figure 5B***). This effect was not the result of a general defect in locomotion or lawn-entry behavior; entry into lawns of non-pathogenic *E. coli* was not affected by AIY inhibition (***Figure 5B***). Inhibition of AVK failed to produce any effect on re-entry behavior (***Figure 5D***). Inhibition of *npr-4*-expressing neurons also increased latency to lawn entry on PA14 (***Figure 5E***). To validate that this effect was due to SIA, we projected patterned light onto P*npr-4:Arch3* animals (***Figure 5F***) to selectively inhibit SIA/SIB. Inhibition of SIA using this method deterred entry of worms onto PA14 lawns (***Figure 5G***), resulting in increased latency in entry (***Figure 5H***). Overall, our results show that two out of the three key neurons identified by compressed sensing were able to elicit a change in lawn entry dynamics through inhibition. Together, our neural imaging and selective inhibition suggested that reduced activity of AIY, SIA, and SIB drives the reduction in lawn occupancy triggered by pathogen exposure.

We next tested whether activation of these neurons would suffice to reverse this behavioral switch. To test this, we optogenetically activated AIY, SIA, and SIB using transgenes expressing the light-gated ion channel channelrhodopsin-2 driven by the two promoters P*ttx-3*::ChR2 and P*npr-4*::ChR2. Transgenic lines were fed ATR and placed on lawns of pathogenic bacteria for 24 hours to induce lawn evacuation. Blue light 467 nm, 1 mW/mm² was then used to activate these neurons and the rate of re-entry onto the lawn was quantified. For both P*ttx-3*::ChR2 (***Figure 5J***) and P*npr-4*::ChR2

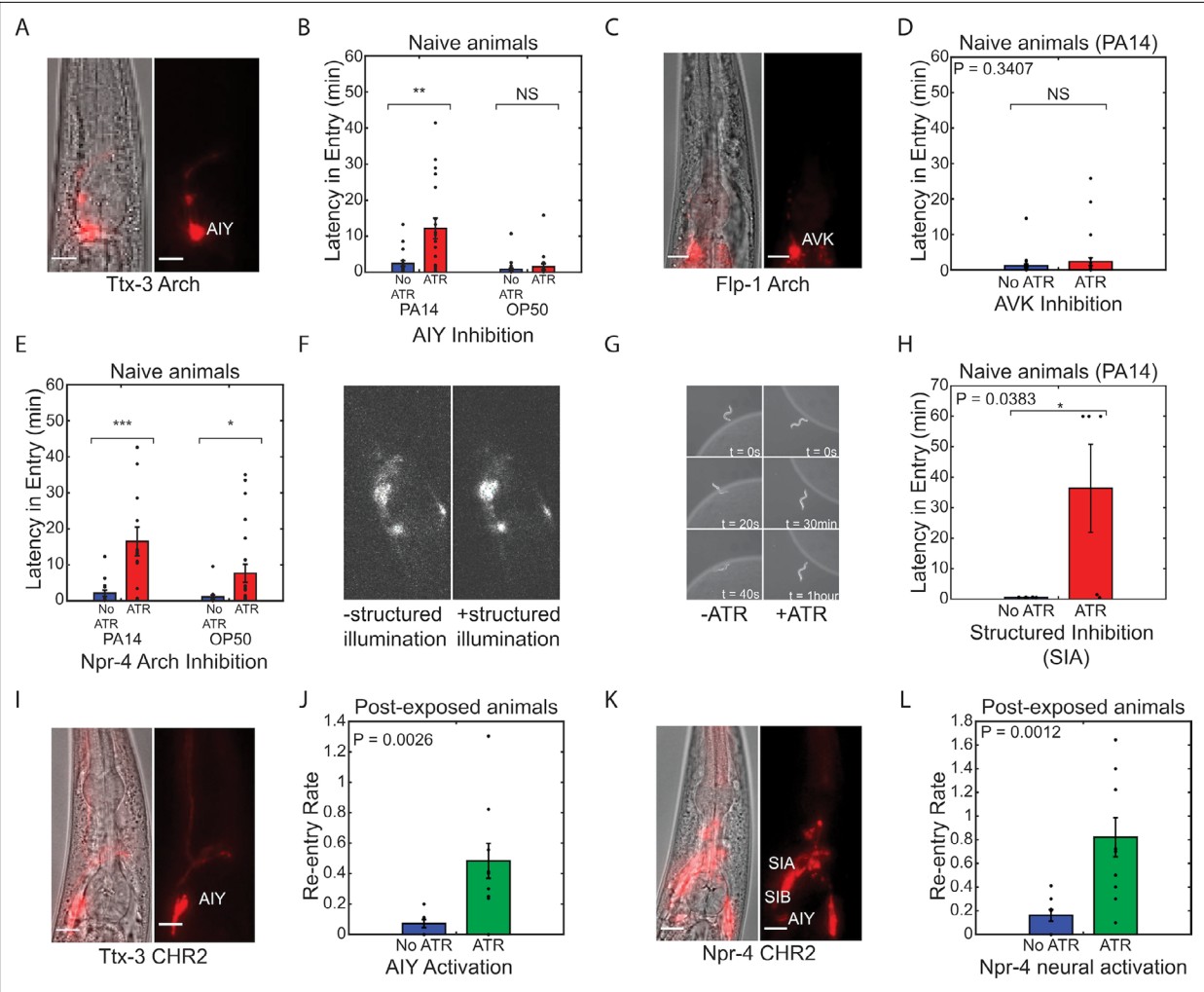

**Figure 5.** Entry onto pathogenic bacteria can be controlled through modulation of key neurons. (**A**) Fluorescent and phase image of P*ttx-3*::ARCH line used for neuron specific AIY inhibition. Scale bar 10 μm. (**B**) Latency in entry (delay in the entry of the animal onto the bacteria lawn upon first contact) of naive worms onto PA14 (n>18) and OP50 (n>17) lawns with (red) and without (blue) AIY neural inhibition. Neural inhibition resulted in significant increases in latency in entry onto PA14, but not OP50. (**C**) Fluorescent and phase image of P*flp-1*::NpHR line used for neuron-specific AVK inhibition. Scale bar 10 μm. (**D**) Latency in entry of worms onto PA14 lawns with (red) and without (blue) AVK neural inhibition (n>27). (**E**) Latency in entry of naive worms onto PA14 (n>11) and OP50 (n>16) with (red) and without (blue) inhibition of *npr-4* expressing neurons. Neural inhibition resulted in significant increases in latency in entry onto both PA14 and OP50. (**F**) Fluorescent images of P*npr-4*::ARCH with (+ATR) and without (-ATR) targeted illumination of SIA/SIB neuron cluster. (**G**) Representative sequence of phase contrast images of a naive worm attempting to enter PA14 bacteria lawn with (+ATR) and without (-ATR) SIA neural inhibition. Time labels denote time after first contact of worm with bacteria lawn. (**H**) Latency in entry of naive worms onto PA14 with (red) and without (blue) SIA neural inhibition. Neural inhibition resulted in a significant increase in latency in entry for PA14. (**I**) Fluorescent and phase image of P*ttx-3*::CHR2 line used for neuron-specific AIY activation. Scale bar corresponds to a length of 10 μm. (**J**) Re-entry rate constant of evacuated worms following 24 hours of exposure (post-exposure) to PA14 lawns with (green) and without (blue) AIY neural activation. Mean and SEM of the rate constant over n=5 samples of 10 worms each (see 'Materials and methods'). (**K**) Fluorescent and phase image of P*npr-4*::CHR2 line used for neural activation. Scale bar corresponds to a length of 10 μm. (**L**) Re-entry rate constant of evacuated worms following 24 hours of exposure to PA14 lawns with (green) and without (blue) *npr-4* neural activation. Mean and SEM of the rate constant over n=5 samples (see 'Materials and methods'). All experiments were performed over a 1-hour time period.

The online version of this article includes the following figure supplement(s) for figure 5:

**Figure supplement 1.** Activation of re-entry neurons fails to impact lawn occupancy.

**Figure supplement 2.** AIY neural activity controls reversal rates.

**Figure supplement 3.** Candidate neuropeptide identification from re-entry neurons.

(*Figure 5L*), activation of these neurons dramatically increased the re-entry rate of experienced animals onto the PA14 lawn. Animals that re-entered the lawn also rapidly exited the lawn. Thus, the increased re-entry rate did not result in sustained increases in lawn occupancy (*Figure 5—figure supplement 1*). These results further illustrate that lawn exit and re-entry are controlled by a distinct set of neurons (*Figure 2C and D*) and demonstrate that both behavioral modules must change in order to evacuate the bacterial lawn.

## Discussion

After experiencing pathogenic bacteria, worms switch their foraging behavior and evacuate the bacterial lawn. We found that this learned pathogen avoidance behavior is driven in part by changes to lawn re-entry behavior. Unlike naive animals, which rapidly re-enter a lawn of pathogen after they exit, animals previously exposed to pathogen dramatically delay re-entry upon encountering the pathogenic bacteria lawn. Such contact-dependent pathogen aversion depends in a graded manner on the extent of pathogen exposure; the latency to lawn re-entry increases with increased time of pathogen exposure. Contact-dependent inhibition of lawn re-entry is a previously unappreciated behavioral response to pathogen exposure revealed by our study. This behavior is distinct from associative olfactory learning and modulation of chemotactic behaviors reported by other studies (*Zhang et al., 2005*, *Ha et al., 2010*). In our study, continuous monitoring of worms exposed to pathogen revealed that animals that leave the pathogen lawn are capable of chemotaxis back to the lawn but stall upon contacting the lawn and do not re-enter. We further found that neurons previously identified as being important in aversive olfactory learning, for example, AIB, RIA, AIZ, and RIM (*Ha et al., 2010*) were not essential for control of lawn re-entry. Our study indicated that the transition in pathogen lawn re-entry behavior occurs through modulation of two key neurons, AIY and SIA, which both decrease their neural activity in response to pathogen exposure. These neurons appear to encode information about the previous pathogen exposure in their neural activity, which in turn influences the worm's behavior when they interact with the pathogenic bacteria. This encoding of experience in neural dynamics is reminiscent of similar processes seen in memory encoding in more complex organisms, such as in the case of the place cells of the hippocampus or the mushroom body.

The capacity of AIY and SIA to inhibit re-entry raises questions about how these neurons might contribute to this behavior. In our previous work, we demonstrated that SIA's neural activity controls speed of locomotion (*Lee et al., 2019*). Reduced SIA neural activity levels might therefore act to inhibit worm movement into the lawn, consistent with the edge stalling behavior seen in worms attempting to re-enter the lawn. Likewise, AIY has been linked to control of reversals and forward locomotion (*Li et al., 2014*). Data we have collected on AIY's neural activity shows a negative correlation with reversal rate (*Figure 5—figure supplement 2A and B*) and inhibition of AIY during locomotion results in increased reversal rate (*Figure 5—figure supplement 2C and D*). Reduced AIY activity may thus result in increased reversals and decreased sustained forward locomotion needed for lawn re-entry.

If contact-dependent lawn aversion represents a modality of pathogen aversion distinct from aversive olfactory learning, what sensory system governs this behavior? While we have yet to establish the signals involved in this process, some hints as to what might be responsible for this aversion can be inferred from looking at synaptic inputs to the neurons that we identified as key for this behavior. SIAs are connected to two sensory neurons that might play a role in driving neural activity changes following pathogen exposure: URX and CEP (*Chen et al., 2006*). URX is a potent regulator of foraging behavior (*Laurent et al., 2015*), is one of the few neurons with contact with the pseudocoelomic fluid of *C. elegans*, and has been previously linked to regulation of metabolic signals and innate immune responses (*Styer et al., 2008*, *Hussey et al., 2018*). CEPs mediate mechanosensory detection of bacteria and potently inhibit locomotion (*Sawin et al., 2000*, *Tanimoto et al., 2016*). By being downstream of these two neurons, SIA might integrate chemosensory and mechanosensory stimuli, two signals known to be important in modulating lawn evacuation from prior studies (*Meisel et al., 2014*, *Chang et al., 2011*). Other neurons that we identified—the AIYs—are part of the olfactory learning circuit and may thus represent a chemotactic component of this contact-dependent pathogen aversion (*Ha et al., 2010*, *Liu et al., 2022b*). In addition, AIY acts as a central hub neuron that is downstream of multiple sensory neurons and may thus also act as an integrator for multiple sensory modalities (*Chalasani et al., 2007*, *Ashida et al., 2019*, *Kuhara et al., 2011*). Interestingly, investigation of molecular modulators of this aversion behavior by looking at neuropeptides that are

highly and uniquely expressed in our neurons of interest reveals a candidate neuropeptide PDF-2 which is highly expressed in AIY (*Figure 5—figure supplement 3A*; *Taylor et al., 2021*). Knocking out PDF-2 increases contact-dependent lawn avoidance (*Figure 5—figure supplement 3B and C*). PDF-2 has been implicated in gut to neuron signaling through the Rictor/TORC2 pathway (*O'Donnell et al., 2018*), suggesting a potential mechanism through which pathogen infection data could be communicated to AIY to influence PDF-2 signaling to modulate behavior.

One remarkable aspect of the neurons that control lawn aversion is the fact that early perturbation of these neurons (within the first 1 to 3 hours of the animal's deposition on the pathogenic lawn) produces long-term changes in pathogen avoidance behavior. Effects of this early neural inhibition could be seen in differences in lawn occupancy 15 hours later, suggesting that early suppression of neural activity patterns has long-term consequences of behavior. How could SIA and AIY produce such long-term effects? One possible explanation is that reduction in AIY and SIA neural activity might serve as an internal cue. The reduction in activity could drive a bistable circuit including AIY and SIA, causing extended suppression of their activity and a long-term change in their neural activity patterns. This bistability could be accomplished through positive autoregulatory feedback. PDF-2, which appears to be involved in bacteria re-entry, might provide such a mechanism to provide this feedback mechanism. Both PDF-2 and its receptor PDFR-1 are expressed in AIY (*Flavell et al., 2013*), providing a potential feedback loop for bistability.

While contact-dependent re-entry represents a novel form of pathogen aversion, it is not the only behavioral transition driving lawn evacuation. Using our compressed sensing-based approach, we were able to rapidly assay the nervous system to not only discover the key neurons controlling entry, but also exit from the pathogen lawn. We find there is little overlap between the sets of neurons controlling these two subbehaviors. Consistent with this, we were able to modulate subbehaviors independently. With the exception of P*dop*-2, lines that showed increases in exit rate upon inhibition did not show changes in lawn re-entry dynamics, and activation of neurons modulating re-entry failed to suppress the exit of worms off the lawn. Together, these results suggest that neural control of lawn evacuation is highly modular, with different sets of neurons governing the individual behavioral transitions needed for lawn evacuation. This modularity in control over net pathogen aversion is similar to that seen in several forms of aversive olfactory learning of *C. elegans* in response to pathogenic bacteria. For example, imprinting of a memory of pathogenic bacteria in larvae stage worms requires distinct sets of neurons for the establishment of the memory and the expression of that memory (*Jin et al., 2016*), while aversive learning in adult worms carries distinct neural circuits for naïve versus learned olfactory preference (*Ha et al., 2010*).

## Materials and methods

### Strains

All lines used in this work are listed in supplementary tables. Lines used for the measurement matrix are in *Supplementary file 1*. Lines used for neural activation and halorhodopsin can be seen in *Supplementary file 2*. Finally, lines used for neural imaging can be seen in *Supplementary file 3*. PDF-2 mutant used was wSR897: nlp-37(tm4780).

### Lawn evacuation assays

PA14 lawn evacuation assay plates were prepared as follows. PA14 was inoculated into LB media and allowed to grow for 15 hours without shaking at 37 °C until the culture reached an OD of 0.1–0.2. 5 uL of this culture was then pipetted onto NGM Agar plates and the colonies were allowed to grow for 18 hours at room temperature. Each colony was surrounded by a ring of filter paper to prevent worms from escaping to the edges of the plate. OP50 lawn evacuation plates were prepared in the same manner as PA14 evacuation plates; however, OP50 bacteria was grown for 18 hours at 37°C with shaking. This longer growth period was used to roughly match the thickness of the PA14 and OP50 colonies.

To begin lawn evacuation, 10 worms were transferred onto the bacteria lawn and given 1 hour to settle. All worms used for this assay were 1-day-old hermaphroditic adults grown on OP50 bacteria. The assay was then imaged at 3 fps for a total of 18 hours to assay lawn avoidance. Occupancy rate

was calculated as (Number of worms on lawn)/(Total number of worms). Worms were counted as being on the lawn if any part of the body was in contact with the lawn.

### Lawn re-entry assay

PA14 bacteria colonies were generated as described for the lawn evacuation assay. Assays were initiated by placing 5–10 worms 5 mm away from the lawn edge, and then imaging the worms at 3 fps for 1 hour. Worms were infected prior to the assay by placing them onto a PA14 lawn evacuation colony for 15 hours. Worms that evacuated the colony over this time period were removed from the plate and used in the assay. Control worms were left on an OP50 lawn during this timeframe.

### Compressed sensing implementation

Compressed sensing was implemented using Lasso regression in Python using the scikit-learn library. Solutions were evaluated over a range of sparsity parameters over 6 orders of magnitude, and chi-squared error of the difference between behavioral predictions and measured behavior was calculated. Solutions shown in *Figure 3* were taken by selecting solutions where chi-squared error began to increase as seen in *Figure 2—figure supplement 1*.

### Optogenetic assay for lawn evacuation

All transgenic lines used for this assay were generated by fusing Archaerhodopsin-3 to the relevant promoters via fusion PCR and then injecting the resulting constructs into worms. Worms used for optogenetic assays were fed on the rhodopsin cofactor all-trans retinal (ATR) for +12 hours before the assay. One-day-old adults were used for all behavioral assays.

Plates were set up as described for the lawn evacuation assays above. Following this, we performed optogenetic inhibition with pulsed (1 second on, 1 second off) 5 mW/mm$^2$ green light for 2 hours using a ScopeLED G250 to activate archaerhodopsin. Assays were imaged for a total of 18 hours either at one frame per 3 minutes for the full assay, or 3 fps to assay entry and exit sub-behaviors.

### Targeted inhibition of neurons during re-entry

Targeted inhibition of SIA and SIB was carried out as described in previous work. Worms were cultured for 12+hours on ATR. Worms were then placed 5 mm away from the edge of a PA14 lawn generated as described above. SIA and SIB were located at the center of the fluorescent pattern P*npr-4::Arch3* line. A circular pattern of light was projected from the DLP projector to selectively target SIA/SIB. Worms were tracked in the frame of view and imaged until they fully entered the PA14 colony or for a total of 1 hour after the worms first contacted the lawn.

### Quantification of entry and exit rate for optogenetic screening

Lawn exit rate was calculated by evaluating the number of exit events during the 2-hour timeframe of neural inhibition. Lawn evacuation movies were analyzed in 1-minute intervals, and any exit events occurring (as defined by a worm completely leaving contact with the bacteria) were noted. The exit events per hour were calculated from this, and statistics were calculated by aggregating exit rates per each experiment replicate (with each replicate consisting of 10 worms on a PA14 lawn). Latency to re-entry was also evaluated by measuring the time from first contact of the worm to the bacteria lawn to time of full entry of the worm onto the lawn. All latency to re-entry analysis was performed by aggregating all works that exited and re-entered the lawn during the measured timeframe. Analysis of statistical significance was performed by carrying out a t-test implemented in Matlab on the phenotype with and without neural inhibition.

### GCAMP imaging and analysis

Calcium imaging was performed using a custom-built microscope as previously described in past work at 15.625 frames per second. Worms were first imaged on an empty NGM plate for 40 minutes. Imaged worms were then transferred onto a colony of PA14 bacteria as used for the lawn evacuation assay and allowed to remain there for 24 hours. Following this, infected worms were imaged again for another 40 minutes.

GCaMP intensity information was extracted using custom software written in MATLAB to extract intensity data given segments encapsulating the neurons of interest. GCaMP imaging data was

compiled for all imaged worms. Worm to worm variability in GCAMP expression was normalized by dividing all data by the bottom 5 percentile of fluorescence intensity in healthy worms. GCaMP data was smoothed over a 6-second window.

### Re-entry with neural activation

PA14 lawn evacuation plates were prepared as described above, and 10 ATR-treated worms expressing channelrhodopsin-2 under the relevant promoters were seeded onto each colony. Channelrhodopsin was activated using 1 mW/mm$^2$ blue light for 1 hour. Re-entry rate (defined as the rate at which worms fully entered the lawn) and contact rate were evaluated and quantified over this time.

### Promoter removal and additions

The robustness of the solutions to removal of promoters was tested as follows. 1–5 promoters were removed from the measurement matrix at random. The resulting measurement matrix and phenotype vector were used to infer neuronal weights for the re-entry phenotype. This process was repeated 200 times for each of the 1–5 promoters, or 1000 times in total for 1–5 promoters. Our results demonstrated robustness of AIY, AVK, SIA, and MI to these removals.

### Robustness of solutions to corruption of measurement matrix

The robustness of the solutions to corruption of the measurement matrix was tested by randomly altering a small fraction of the measurement matrix and re-inferring the neural contributions. 10% of the non-zero entries in the measurement matrix were altered to a value between 0 and 0.5. 1000 such corrupted measurement matrices were generated and solutions to each corrupted matrix were inferred via Lasso regression. Our results demonstrated that corruption generally did not result in alteration of the inferred neurons, with AIY, AVK, MI, and SIA robustly being inferred even with matrix corruption.

### Calculation of inverse participation ratio and relative expression

Single-cell RNA seq data from Taylor et al. was used to calculate the average expression of each neuropeptide within each neuron type. The relative expression of each of the neuropeptides in each neuron was calculated by dividing expression in each neuron by the maximum expression over all neurons. Inverse participation ratio (IPR) was calculated as

$$IPR = \frac{1}{\sum_i \left( \frac{E_i}{\sum_i E_i} \right)^2}$$

where $E_i$ is the expression in the $i$th neuron.

## Acknowledgements

We would like to thank members of the Ringstad and Ramathan lab for their feedback and advice on the manuscript. This work is supported by 5R01NS117908-03 (SR, NR).

## Additional information

### Funding

| Funder | Grant reference number | Author |
| --- | --- | --- |
| National Institutes of Health | 5R01NS117908-03 | Niels Ringstad<br>Sharad Ramanathan |

The funders had no role in study design, data collection and interpretation, or the decision to submit the work for publication.

## Author contributions
Timothy Hallacy, Conceptualization, Data curation, Software, Formal analysis, Validation, Investigation, Visualization, Methodology, Writing – original draft; Abdullah Yonar, Conceptualization, Formal analysis, Investigation, Visualization, Methodology; Niels Ringstad, Supervision, Methodology, Writing – review and editing; Sharad Ramanathan, Conceptualization, Supervision, Investigation, Methodology, Writing – original draft, Project administration, Writing – review and editing

## Author ORCIDs
Timothy Hallacy ⓘ https://orcid.org/0000-0002-0069-1409
Niels Ringstad ⓘ https://orcid.org/0000-0002-8679-2269
Sharad Ramanathan ⓘ https://orcid.org/0000-0001-9445-1248

Reviewer #1 (Public review): https://doi.org/10.7554/eLife.97340.3.sa1
Reviewer #2 (Public review): https://doi.org/10.7554/eLife.97340.3.sa2
Author response https://doi.org/10.7554/eLife.97340.3.sa3

## Additional files

### Supplementary files
Supplementary file 1. Archaerhodopsin lines that constitute the measurement matrix.

Supplementary file 2. Channelrhodopsin and halorhodopsin lines.

Supplementary file 3. GCaMP lines.

MDAR checklist

### Data availability
Processed behavioral, imaging, and screening data, together with analysis code, are openly available at https://github.com/TimHVAnalytica/C-elegans-pathogen-avoidance-data (copy archived at *Hallacy, 2025*).

The following previously published dataset was used:

| Author(s) | Year | Dataset title | Dataset URL | Database and Identifier |
| --- | --- | --- | --- | --- |
| Taylor SR, Miller DM | 2019 | Molecular topography of an entire nervous system | https://www.ncbi.nlm.nih.gov/geo/query/acc.cgi?acc=GSE136049 | NCBI Gene Expression Omnibus, GSE136049 |

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
