## [Editor Report · eLife Assessment]

This **important** study describes a neural circuit contributing to two behavioral processes affecting pathogen avoidance in the nematode *C. elegans*. The method used to identify specific contributing neurons is innovative, and the experimental evidence supporting the major claims is **solid**. This study will be of interest to neuroscientists studying behavior, in particular in *C. elegans*.

---

## [Referee Report · Reviewer #1 (Public review)]

This study identifies two behavioral processes that underlie learned pathogen avoidance behavior in *C. elegans*: exiting and re-entry of pathogenic bacterial lawns. Long-term behavioral tracking indicates that animals increase the prevalence of both behaviors over long-term exposure to the pathogen *Pseudomonas aeruginosa*. Using an optogenetic silencing screen, the authors identify groups of neurons, whose activity regulates lawn occupancy. Surprisingly, they find that optogenetic inhibition of neurons during only the first two hours of pathogen exposure can establish subsequent long-term changes in pathogen aversion. By leveraging a compressed sensing approach, the authors define a set of neurons involved in either lawn exit or lawn re-entry behavior using a constrained set of transgenic lines that drive Arch-3 expression in overlapping groups of neurons. They then measure the calcium activity of the candidate neurons involved in lawn re-entry in freely moving animals using GCaMP, and observe a reduction in their neural activity after exposure to pathogen. Optogenetic inhibition of AIY and SIA neurons during acute pathogen exposure in naïve animals delays lawn entry whereas activating these neurons in animals previously exposed to pathogen enhances lawn entry, albeit transiently.

This work is missing experiments and analyses that are necessary to substantiate their claims. Although the authors convincingly show that neuronal inhibition experiments during pathogen exposure reveal separable groups of neurons controlling pathogenic lawn exiting and re-entry, their methods to validate these results at single neuron cell-type resolution lack rigor.

In Figure 4, the authors claim that the reduction in calcium activity in cells of interest following pathogen exposure encodes pathogen experience. However, they make no effort to correlate the observed decreased activity with concomitant shifts in increased immobility (decreased forward locomotion) or the increased age of the worms since pathogen exposure began (24 hours have elapsed), either of which could easily explain these results. A better comparison would be between age-matched naive animals and animals exposed to pathogen. More to the point, we are interested in the involvement of these neurons' activity patterns with the behavioral motifs associated with lawn exits and re-entries, so examining these activity patterns in the absence of any pathogen before or after long-term pathogen exposure yields little insight into their relevant signaling roles. To substantiate the authors' claims, a better experiment would measure these neurons' calcium activity during lawn exits and re-entries in naive and post-exposed age-matched worms.

In Figure 5, the authors attempt to show that manipulating AIY and SIA/SIB neuronal activity controls pathogenic lawn re-entry behavior. Although they show that inhibiting these neurons in naive animals increases latency to enter pathogenic lawns, they never test the effect of neuronal inhibition in post-exposed animals. Instead they activate these neurons using channelrhodopsin, whereby they observe an increase in lawn entry and exit behavior, indicative of high forward locomotion speed. Although suggestive, neither of these experiments prove these neurons' involvement in pathogenic lawn re-entry behavior following pathogen exposure. To rigorously test the hypothesis that AIY and SIA/SIB neurons are required to sustain higher latency to lawn re-entry following pathogen exposure, the authors should perform neuronal inhibition experiments in post-pathogen-exposed animals as well and compare the results. The interpretation of this figure is further complicated by the fact that Npr-4::ChR2 animals express ChR2 in AIY in addition to SIA/SIB neurons: experiments that calculated lawn re-entry rates in Npr-4::ChR2 activation in post-exposed animals may include the known effect of stimulating AIY alone (Fig. 5J) since no discernible attempt at structured illumination to limit excitation to SIA/SIB neurons was made in these animals (Fig. 5 K, L).

This work raises the interesting possibility that different sets of neurons control lawn exit and lawn re-entry behaviors following pathogen exposure. However, the authors never directly test this claim. To rigorously show this, the authors would need to show that lawn-exit promoting neurons (CEPs, HSNs, RIAs, RIDs, SIAs) are dispensable for lawn re-entry behavior and that lawn re-entry promoting neurons (AVK, SIA, AIY, MI) are dispensable for lawn exit behavior in pathogen-exposed animals. The authors identify AVK neurons as important for modulating lawn re-entry behavior by brief inhibition at the start of pathogen exposure but fail to find that these neurons are required for increased latency to re-entry in naïve animals (Fig. 5D). Recent work from Marquina-Solis et al (2024) shows that chronic silencing of these neurons delays pathogen lawn leaving, due to impaired release of flp-1 neuropeptide. Authors may wish to connect their work more closely with the existing literature by investigating the behavioral process by which AVK contributes to lawn evacuation.

---

## [Referee Report · Reviewer #2 (Public review)]

In this manuscript, Hallacy et al. used a compressed sensing-based optogenetic screening method to investigate the crucial neurons that regulate pathogenic avoidance behavior in *C. elegans*. They further substantiate their findings using complementary optogenetic activation and imaging techniques to confirm the roles of the key neurons identified through extensive screening efforts. Notably, they identified AIY and SIA as pivotal neurons in the dynamic process of pathogenic avoidance. Their significant discovery is the delayed or stalled reentry process, which drives avoidance behavior; to my knowledge, this dynamic has not been previously documented. Additionally, the successful integration of quantitative optogenetic tools and compressed sensing algorithms is noteworthy, demonstrating the potential for obtaining highly quantitative data from the *C. elegans* nervous system. This approach is quite rare in this field, yet it represents a promising direction for studying this simple nervous system.

However, the paper's main weakness lies in its lack of a detailed mechanism explaining how the delayed reentry process directly influences the actual locomotor output that results in avoidance. The term 'delayed reentry' is used as a dynamic metric for quantifying the screening, yet the causal link between this metric and the mechanistic output remains unclear. Despite this, the study is well-structured, with comprehensive control experiments, and is very well constructed.

Comments on revisions:

The authors have addressed all my concerns and suggestions. They particularly further clarified the AIY's role in navigation by providing a new figure. They also provided supplementary videos representing the re-entry process.

---

## [Author Response]

The following is the authors’ response to the original reviews.

**Reviewer 1:**

We thank the reviewer for their comments and suggestions. We have made several edits to the paper to address these comments, including the addition of several new control experiments, corrections to mislabeled figures in Fig 2, and other additions to improve the clarity of several figures.

This work is missing several controls that are necessary to substantiate their claims. My most important concern is that the optogenetic screen for neurons that alter pathogenic lawn occupancy does not have an accompanying control on non-pathogenic OP50 bacteria. Hence, it remains unclear whether these neuronal inhibition experiments lead to pathogen-specific or generalized lawn-leaving alterations. For strains that show statistical differences between - and + ATR conditions, the authors should perform follow-up validation experiments on non-pathogenic OP50 lawns to ensure that the observed effect is PA14-specific. Similarly, neuronal inhibition experiments in Figures 5E and H are only performed with naïve animals on PA14 - we need to see the latency to re-entry on OP50 as well, to make general conclusions about these neurons' role in pathogen-specific avoidance.

We have added data from new control experiments to Fig. S1 (subfigures B, C) for both exit and re-entry dynamics on OP50. We find that inhibition of neurons produces different effects on both lawn entry and exit on PA14 compared to OP50. We observed that inhibition of neurons failed to change the re-entry dynamics for any of the lines which showed delayed latency to re-entry on PA14. Our results suggest that the neural control of re-entry dynamics we see are PA14 specific.

My second major concern is regarding the calcium imaging experiments of candidate neurons involved in lawn re-entry behavior. Although the data shows that AIY, AVK, and SIA/SIB neurons all show reduced activity following pathogen exposure, the authors do not relate these activity changes to changes in behavior. Given the well-established links between these cells and forward locomotion, it is essential to not only report differences in activity but also in the relationship between this activity and locomotory behavior. If animals are paused outside of the pathogen lawn, these neurons may show low activity simply because the animals are not moving forward. Other forward-modulated neurons may also show this pattern of reduced activity if the animals remain paused. Given that the authors have recorded neural activity before and after contact with pathogenic bacteria in freely moving animals, they should also provide an analysis of the relationship between proximity to the lawn and the activity of these neurons.

In response, we added an additional supplementary figure S7 to illustrate the role of each neuron in navigational control and added text to the discussion to better explain the role of each neuron type in the regulation of re-entry, in light of our previously published work on SIA in speed control.

This work is missing methodological descriptions that are necessary for the correct interpretation of the results shown here. Figure 2 suggests that the determination of statistical significance across the optogenetic inhibition screen will be found in the Methods, but this information is not to be found there. At various points in the text, authors refer to "exit rate", "rate constant", and "entry rate". These metrics seem derived from an averaged measurement across many individual animals in one lawn evacuation assay plate. However "latency to re-entry" is only defined on a per-animal basis in the lawn re-exposure assay. These differences should be clearly stated in the methods section to avoid confusion and to ensure that statistics are computed correctly.

Additional details have been added to the methods section to provide more in depth information on the statistical analysis performed. In brief, the latency to re-entry is calculated in the same way across all assays – re-entry events across replicate experiments for a given experimental condition are aggregated together and used to calculate relevant statistics.

This work also contains mislabeled graphs and incorrect correspondence with the text, which make it difficult to follow the authors 'claims. The text suggests that Pdop-2::Arch3 and Pmpz-1::Arch3 show increased exit rates, whereas Figure 2 shows that Pflp-4::Arch3 but not Pmpz-1::Arch3 has increased exit rate. The authors should also make a greater effort to correctly and clearly label which type of behavioral experiment is used to generate each figure and describe the differences in experimental design in the main text, figure legends, and methods. Figure 2E depicts trajectories of animals leaving a lawn over a 2.5-minute interval but it is unclear when this time window occurs within the 18-hour lawn leaving assay. Likewise, Figure 2H depicts a 30-minute time window which has an unclear relationship to the overall time course of lawn leaving. This figure legend is also mislabeled as "Infected/Healthy", whereas it should be labeled "-/+ ATR".In Figures 2C and F, the x-axis labels are in a different order, making it difficult to compare between the 2 plots. Promoter names should be italicized. What does the red ring mean in Figure 2A? Figure 2 legend incorrectly states that four lines showed statistically significant changes for the Exist rate constant - only 2 lines are significant according to the figure.

We thank the reviewer for identifying this embarrassing error. Figure 2C and F were flipped, and we have corrected this, we are sorry for the error. Promoter names have been italicized, and we have added additional text in the captions that the red ring is a ring light for background illumination of the worms. In addition, we have corrected the error in the figure legends from “Infected/Healthy” to “+/- ATR”.

Lines in figure 2C and 2F are ordered by significance rather than keeping the same order in both. Majority feedback from colleagues suggested that this ordering was preferred.

This work raises the interesting possibility that different sets of neurons control lawn exit and lawn re-entry behaviors following pathogen exposure. However, the authors never directly test this claim. To rigorously show this, the authors would need to show that lawn-exit-promoting neurons (CEPs, HSNs, RIAs, RIDs, SIAs) are dispensable for lawn re-entry behavior and that lawn re-entry promoting neurons (AVK, SIA, AIY, MI) are dispensable for lawn exit behavior in pathogen-exposed animals.

We agree with the reviewer’s comments that there is insufficient evidence to show a complete decoupling of lawn exit and lawn re-entry. However, we note that our screen results show that only 1 line (dop-2) shows changes in both exit and re-entry dynamics upon neural inhibition (Fig. 2). This seems to suggest that at least some degree of neural control of re-entry is decoupled from exit.

Please label graph axes with units in Figure 1 - instead of "Exit Rate" make it #exits per worm per hour, and make it more clear that Figures 1C and E have a different kind of assay than Figures 1A, B and D. There should be more consistency between the meaning of "pre/post" and "naive/infected/healthy" - and how many hours constitutes post.

We have edited Figure 1 and made additions to the captions of figure 1 to make both points clearer. We have also standardized our language for subsequent figures (such as figure 5) to provide less ambiguity in pre/post and naïve/infected/healthy.

Figure 5 - it should be made more clear when the stimulation/inhibition occurred in these experiments and how long they were recorded/analyzed.

We have added additional details to the figure captions to make it clearer when the data was collected.

Workspaces and code have been added under a data availability section in the manuscript.

**Reviewer 2:**
However, the paper's main weakness lies in its lack of a detailed mechanism explaining how the delayed reentry process directly influences the actual locomotor output that results in avoidance. The term 'delayed reentry' is used as a dynamic metric for quantifying the screening, yet the causal link between this metric and the mechanistic output remains unclear. Despite this, the study is well-structured, with comprehensive control experiments, and is very well constructed.

We thank the reviewer for their comments and suggestions. We have added additional data and details to our work to cover these weaknesses, as can be seen in our responses to the suggestions below.

(1) A key issue in the manuscript is the mechanistic link between the delayed process and locomotor output. AIY is identified as a crucial neuron in this process, but the specifics of how AIY influences this delay are not clear. For instance, does AIY decrease the reversal rate, causing animals to get into long-range search when they leave the bacterial lawn? Is there any relationship between pdf-2 expression and reversal rates? Given that AIY typically promotes long-range motion when activated, the suppression of this function and its implications on motion warrants further clarification.

We have included additional data to explain how AIY might be able to regulate lawn entry behaviors and have added more to the discussion to explain how neural suppression might lead to changes in the behavior (new figure S7). Both AIY and SIA dynamics have been linked to worm navigation. In previous work (Lee 2019), we have demonstrated that SIA can control locomotory speed. Inhibition of SIA decreases locomotory speed, and as a result may serve to drive the increased latency of re-entry.

AIY’s role in navigation has been previously established (Zhaoyu 2014), but we have added an additional supplementary figure and edited our discussion to further illustrate this point. As can be seen in the new figure S7, AIY neural activity undergoes a transition after removal from a bacterial lawn, going from low activity to high activity. This activity increase is correlated with a transition from a high reversal rate local search state to a long range search state characterized by longer runs. Inhibition of AIY during this long range search state increased the reversal rate resulting in a higher rate of re-orientations. This might serve as a part of the mechanistic explanation for AIY’s role in preventing lawn re-entry, as inhibition dramatically increased the rate of re-orientation, preventing worms from making directed runs into the bacterial lawn. However, there is an additional effect of the inhibition of AIY, not seen during food search. Inhibition of AIY in the context of a pathogenic bacterial lawn leads to stalling at the edge. Therefore, re-entry AIY could have an additional role in governing the animals movement, post exposure, upon contact with a pathogenic lawn.

(2) I recommend including supplementary videos to visually demonstrate the process. These videos might help others identify aspects of the mechanism that are currently missing or unclear in the text.(4) The authors mention that the worms "left the lawn," but the images suggest that the worms do not stray far and remain around the perimeter. Providing videos could help clarify this observation and strengthen the argument by visually connecting these points

Additional supplementary videos (1-3) taken at several stages of lawn evacuation have been added to visually demonstrate the process.

(3) Regarding the control experiments (Figure 1E-G), the manuscript describes testing animals picked from a PA14-seeded plate and retesting them on different plates. It's crucial to clarify the differences between these plates. Specifically, the region just outside the lawn should be considered, as it is not empty and worms can spread bacteria around. Testing animals on a new plate with a pristine proximity region might introduce variables that affect their behavior.

We have reworded the paper to make it clearer that these new conditions on a fresh PA14 lawn represent a different type of assay from the lawn evacuation assay. Fresh PA14 plates will indeed have a pristine proximity region compared to plates where the worms have spread the bacteria.

These experiments were done to test if the evacuation effect is purely due to aversive signals left on the lawn or attractive signals left outside of the lawn. Given that worms are known to be able to leave compounds such as ascarosides to communicate with each other, we wanted to test that this lawn re-entry defect was not simply the result of deposited pheromones. Without any other method to remove such compounds, we relied on using fresh PA14 lawns instead to test this. We have updated the manuscript to clarify this point.

(5) The manuscript notes that the PA14 strain was grown without shaking. Typically, growing this strain without agitation leads to biofilm formation. Clarifying whether there is a link between biofilm formation and avoidance behavior would add depth to the understanding of the experimental conditions and their impact on the observed behaviors.

As the reviewer has noted, growth of PA14 without shaking might indeed lead to biofilm formation. This does represent a legitimate concern, as evidence from previous work has suggested that biofilm formation could be linked to pathogen avoidance as worms make use of mechanosensation to avoid pathogenic bacteria (Chang et al. 2011). However, we do not observe substantial formation of biofilm in our cultured bacteria, likely since our growth time might be insufficient for sufficient biofilm formation to occur. We also note that our evacuation dynamics appear to be of similar timescale to results reported in previous work which used different growth conditions. As such, we believe that our growth conditions thus represent similar conditions as to those historically used in the lawn evacuation literature.

**Reviewer 3:**
Weaknesses:My only concern is that the authors should be more careful about describing their "compressed sensing-based approach". Authors often cite their previous Nature Methods paper, but should explain more because this method is critical for this manuscript. Also, this analysis is based on the hypothesis that only a small number of neurons are responsible for a given behavior. Authors should explain more about how to determine scarcity parameters, for example.

We have added more details to our paper outlining some of the details involved in our compressed sensing approach. We go into more detail about how we chose sparsity parameters and note that our discovered neurons for re-entry appear to be robust over choice of sparsity parameters. These additional details can be found in both the paper body and the methods section.

Line 45: This paragraph tries to mention that there should be "small sets of neurons" that can play key roles in integrating previous information to influence subsequent behavior. Is it valid as an assumption in the nervous systems?

We want to clarify that what is important is not that there are ‘small sets of neurons’, but rather that these key neurons make up a small fraction of the total number of neurons in the nervous system. More correctly: the compressed sensing approach identifies information bottlenecks in the neural circuits, and the assumption is that the number of neurons in these bottlenecks are small. This is the underlying sparsity assumption being made here that allows us to utilize a compressed sensing based approach to identify these neurons. We have reworded this section to make it clear that what is important is not that the total number of neurons is small, but that they must be a small fraction of the total number of neurons in the nervous system.

Line 125: "These approaches…" Authors repeatedly mentioned this statement to emphasize that their compressed sensing-based approach is the best choice. Are you really sure?

We agree that there are several approaches that might allow for faster screening of the nervous system. For example, many studies approach the problem by looking at neurons with synapses onto a neuron already known to be implicated in the behavior or find neurons that express a key gene known to regulate the behavior of interest. These approaches utilize prior information to greatly reduce the pool of candidate neurons needed to be screened.

In the absence of such prior information, we believe that our compressed sensing based approach allows a rapid way to perform an unbiased interrogation of the entire nervous system to identify key neurons at bottlenecks of neural circuits. Once these key neurons are identified, neurons upstream and downstream of these key neurons can be investigated in the future. This approach gives us the added advantage of being able to identify neurons that do not connect to neurons that are already implicated in the behavior, or that don’t have clear genetic signatures in the behavior of interest. Our approach further allows for screening of neurons with no clear single genetic marker without the next to utilize intersectional genetic strategies. We should not use the phrase “best choice” which might not be justified. We have reworded these statements, and we believe that compressed sensing based methods provide a complementary approach to those in the literature.

Line 42: If authors refer to mushroom bodies and human hippocampus in relation to the significance of their work, authors should go back to these references in the Discussion and explain how their work is important.

We thank the reviewer for this feedback, and we have added to our discussion to expand upon these points.

Line 151: "the accelerated pathogen avoidance" Accelerated pathogen avoidance does not necessarily indicate the existence of the neural mechanism that inhibits the association of pathogenicity with microbe-specific cues (during early stages: first two hours).

We agree with the reviewer’s statements that these results alone do not indicate the presence of an early avoidance mechanism. Other evidence for early avoidance mechanisms exists as seen in two choice assay experiments (Zhang 2005), and our results do seem to support this. However, we agree that early neural inhibition is insufficient evidence towards such a mechanism. We have thus removed this statement for accuracy.